# Comparative Analysis on the Effect of Surface Reflectance for Laser 3D Scanner Calibrator

**DOI:** 10.3390/mi13101607

**Published:** 2022-09-27

**Authors:** Jia Ou, Tingfa Xu, Xiaochuan Gan, Xuejun He, Yan Li, Jiansu Qu, Wei Zhang, Cunliang Cai

**Affiliations:** 1School of Optics and Photonics, Beijing Institute of Technology, Beijing 100081, China; 2Beijing Changcheng Institute of Metrology and Measurement, Beijing 100095, China; 3School of Opto-Electronic Engineering, Changchun University of Science and Technology, Changchun 130022, China

**Keywords:** laser scanning instrument, 3D scanner calibrator, surface reflectance, measurement accuracy

## Abstract

The calibrator is one of the most important factors in the calibration of various laser 3D scanning instruments. The requirements for the diffuse reflection surface are emphasized in many national standards. In this study, spherical calibrator and plane calibrator comparative measurement experiments were carried out. The black ceramic standard sphere, white ceramic standard sphere, metal standard sphere, metal standard plane, and white ceramic standard plane were used to test the laser 3D scanner. In the spherical calibrator comparative measurement experiments, the results indicate that the RMS of the white ceramic spherical calibrator with a reflectance of approximately 60% is 10 times that of the metal spherical calibrator with the reflectance of approximately 15%, and the RMS of the black ceramic spherical calibrator with reflectance of approximately 11% is of the same order as the metal spherical calibrator. In the plane calibrators comparative measurement experiments, the RMS of the flatness measurement is 0.077 mm for the metal plane calibrator with a reflectance of 15%, and 2.915 mm for ceramic plane calibrator with a reflectance of 60%. The results show that when the optimal measurement distance and incident angle are selected, the reflectance of the calibrator has a great effect on the measurement results, regardless of the outlines or profiles. Based on the experiments, it is recommended to use the spherical calibrator or the standard plane with a reflectance of around 18% as the standard, which can obtain reasonable results. In addition, it is necessary to clearly provide the material category and surface reflectance information of the standard when calibrating the scanner according to the measurement standard.

## 1. Introduction

Non-contact measurement technology is widely used, especially for the large-scale measurement of workpieces, such as propeller blades [1], antenna reflectors [2], cabin capacity [3,4], vertical metal containers, etc. This technology can realize long-distance, high-precision and fast measurement for large-scale objects, which avoids interfacial contact and possible deformation in the measurement. Digital photogrammetry and scanning measurement methods are two important components in non-contact measurement. Digital photogrammetry mainly refers to the measurement technology that utilizes digital cameras to obtain the images of the objects. Afterwards, these images were processed with further digital processing to obtain high-precision and three-dimensional (3D) contours of the measured objects’ surface [5]. Generally, cooperative targets, such as marked points (reflective points or non-reflective points) and coding points need to be pasted on the surface of the measured object, which can be extracted by imaging processing software to measure the object.

Some laser 3D scanners generally do not need cooperative targets. The instrument projects visible features to the measured object. The common features include point, line [6], grating [7], speckle [8], and other forms. For scanning measurement, the measuring light irradiates the surface of the measured object. The measured surface must be able to reflect a certain amount of light to ensure that the object surface contour can be obtained by reflected light. After being reflected by the measured surface, the imaging detector can precisely catch the light spot, and then the position of the light spot is further processed to finally achieve the result.

Three-dimensional scanning measurement instruments usually need to calibrate their accuracy before measurement. The common calibration method is to use a standard sphere with a certain diameter or a standard flat plane with a certain dimension. In practice, it is noted that the surface reflectance of the standard sphere/plane has great influence on the accuracy of the measurement results. In order to reduce the effect of uneven reflection, powder spraying pretreatment is usually required in three-dimensional scanning operation, which can produce a dull and sub-white coating on the workpiece. The spraying layer reduces the uneven reflectance of light and realizes much better scanning conditions. Spraying different powder has different effects on the surface reflectance. For instance, the size of titanium spraying powder is approximately 1~2 um, and another kind of spraying powder size of DPT-5 is approximately 5~6 um [9,10]. Although spraying solves the issue of surface uneven reflectance, it alters the original surface roughness, as well as the size of the object. The calibrator for the measurement, whether it is a standard sphere or a standard plane, has a strict geometrical tolerance and smooth surface roughness, which cannot be demonstrated with spraying powder. Therefore, it is necessary to study the effect of surface reflectance of the standard sphere or plane on the measurement results, looking for an appropriate surface reflection for direct application without spraying.

To address this issue, some related parameters, such as color and light source, that might affect the scanning measurement results have been studied [11,12,13]. In some typical applications, such as the influence of reflectance in the laser 3D scanning of weld grooves proposed by Andrej Cibicik et al. [14], the change in reflectance produced more noise point cloud data. It is clear that when the light source is projected onto the metal surface, it may produce high-intensity reflected light and light pollution, which affects the measurement results. This article also provided methods to reduce the influence of noise. David A. Gonzalez et al. [15] mentioned the influence of surface reflectance and incident angle on surface measurement in the process of studying a narrow-angle flash nano-surface with reflectance suppression, and studied the method to reduce the influence of reflectance on measurement from the perspective of the surface micro–nano structure. Tania Das et al. [16] also pointed out that incident angle and reflectance had a great impact on the measurement results when using polarized light to measure the surface topography of metal, and studied the method of using polarized light to detect the surface contour of a metal plane. In many studies, the influence of reflectance on the measurement of the laser scanning method has been found, but few studies have studied the influence of reflectance on the measurement accuracy from the perspective of the calibration of a laser scanner.

In this study, using a variety of materials with different surface reflectances, combined with scanning measurement tests, the effect of surface reflectance on the scanning measurement result is investigated. It is found that reflectivity has a great influence on the calibration results, but the related standards VDI/VDE 2634 and JJF 1951–2021 are not mentioned. This article first puts forward different reflectivity of the same 3d scanning device scans have different effect, this conclusion can not only improve existing calibration specifications, can also for other scanning measurement calibration equipment research and development and to provide the reference.

## 2. Materials and Methods

### 2.1. Principle of Laser Scanning Measurement

According to the number of scanning points at one time, the instruments can be classified into point scanning, linear scanning, and plane scanning. The point scanning method can only measure one point at one time because of the limitation of distance measurement theory. Based on the principle of spherical coordinates, as shown in Figure 1, the position of point *P* (*x*, *y*, *z*) in the spherical coordinate can be obtained based on the radius *r*, azimuth *α*, and elevation *θ*. The value of *r* can be obtained by collecting the duration of light travelling between the instrument and the object, and the value of *α* and *θ* can be obtained by angle encoders in the instrument. A point scanning system is commonly found in MV330 LASER RADAR produced by Nikon, Tokyo, Japan, which uses a ranging technology of frequency modulation coherent LIDAR technology [17], and an ATS600 absolute tracker and various ground-based laser scanners produced by Leica, Wetzlar, Germany.
(1)$$\left\{\begin{array}{l} x_{P}=r\sin\theta \\ y_{P}=r\cos\theta \cos\alpha \\ z_{P}=r\cos\theta \sin\alpha \end{array}\right.$$

The line scanning method projects a single laser line or multiple laser lines to the measured surface, as shown in Figure 2. Numerous points of measurement along the line can be collected at one time. The number of points along the line scanning is related to the hardware of the scanning system [18]. The line scanning systems are commonly found in the LJ-G5000 series laser 2D/3D high-precision measurement systems produced by Keyence, Osaka, Japan.

The area scanning system [11] projects grating, fringe, speckle, and other patterns onto the measured surface. In general, more measurement points can be obtained at one time than line scanning. The area scanning system makes use of stereo vision and structured light [19,20], as shown in Figure 3. The structured light is projected onto the surface of the measured object through the transmission of a light source and a 3D image fitting the shape of the measured object is formed on its surface. The 3D image is captured by two cameras to obtain the distorted image modulated by the object’s spatial dimension. The 3D point cloud information of the object surface is obtained for further image processing, and the image point cloud is divided into fields by software [21,22]. The 3D geometric features are synthesized to re-construct the measured surface. The representative instruments are the MetroSCAN series scanners produced by Creaform, Lévis, PQ, Canada and TrackScan-P series scanners produced by SCANTECH company, Beijing, China.

No matter what kind of laser scanner, its basic principle is that the receiving system receives the laser reflected light irradiated on the surface of the measured object. If the energy of reflected light is too low, the receiving signal contains much comparable noise from the surroundings. On the contrary, if the reflected light is over-strong, the receiving system may be saturated, resulting in the wrong data. Therefore, if the laser scanner is calibrated, it is necessary to study the differences between standards with different reflectance and find a suitable reflectance standard sphere or standard plate.

### 2.2. Effect of Incident Angle and Reflectance on 3D Scanning

The laser scanner collects the 3D information of the objects through the point cloud data [23]. Theoretically, the difference can be achieved by comparing the measured data of the scanner with the known coordinate points. However, in practical application, it is difficult to have a sufficiently small object as the target point. Therefore, using the standard sphere and standard plane is a common method to calibrate the scanning system. The calibrated 3D scanning system constructs the point cloud data of the standard sphere to obtain the diameter of the standard sphere, and then compares the data with the known radius of the standard sphere. Similarly, the data of the standard plane obtained by the calibrated scanning system can also be compared with the known data to judge the measurement accuracy of the 3D scanner system. In this way, whether it is a standard sphere or a standard plane, the reflection characteristics will affect the accuracy of the measurement results.

For different materials, the change trend between the laser incidence angle and the reflection intensity is similar. The reflection intensity can be determined by the relationship between the laser incidence angle and the reflection intensity [24].

The surface reflectance of the measured object is a key parameter related to the measurement ability of the laser scanner. The main factors affecting the reflectance of object surface are object surface color and surface type (matte, highlight, smooth, rough, etc.,). The surface reflectance of Kodak white material is generally defined as 100%, and that of Kodak gray material as 18% [25]. In order to reduce the influence of reflectance on the measurement results, the common method is to find a suitable incidence angle and provide a standard sphere or standard plane with appropriate reflectance.

## 3. Experiment

In order to research the effect of reflectance on the calibration result of the scanner, a white ceramic ball with higher reflectance, black ceramic ball with lower reflectance, and metal ball with lower reflectance were used as standard spheres and their surface reflectances were determined.

Before the test is carried out, we need to clean the surface of the standard instrument under test. The accuracy of the measurement results is ensured by removing surface oil or cutting fluid residue through cleaning operation. First, two standard spheres with different reflectance of ceramic materials were tested and the radii of the two balls measured by the scanner were compared. The three standard spheres were then placed together to compare the material’s effect on the scans. In order to exclude the influence of incident angle on the measurement results, we made a low reflectance metal plane and high-reflectance ceramic plane as the standard plane, to verify the influence of reflectance on the measurement results.

The measured standard ball is placed on the marble platform, and the lidar distance to the standard ball is about 3 m. The measurement process is described as follows: 1. lidar startup preheat 1 h; 2. using SA software, measurement mode is “measurement mode”; 3. setting the scanning measuring range to cover the standard ball; 4. starting scanning. 5. exporting the point cloud file to PolyWorks software; 6. delete obvious outliers; 7. triangulating the point cloud to facilitate observation; 8. setting the fitting parameters and obtain the results by fitting features.

### 3.1. Reflectance Measurement of the Standard Spheres

In accordance with the measurement standard JJF 1032–2005, the definition of reflectance is given as follows: under the specified conditions of the spectral composition, polarization state, and geometric distribution of the incident radiation, it is the ratio of the reflected radiant flux to the incident flux. The JJF 1232–2009 calibration specification for a reflectance meter is used to measure the diffuse reflectance of paint, coating, ceramics, and other materials. The reflectance of most common materials can be obtained by such instruments.

In this study, a spectrophotometer (produced by SPECTRO, Germany, Kleve) is used to measure the surface reflectance. The colorimeter adopts a D/8 geometric optical structure [26,27,28], which is a diffuse reflection illumination. The 8° detection method with a 2°/10° standard observer angle and D65 standard light source is used for measurement. Specular component included (SCI) is a measurement mode that includes specular reflection light, and SCE (specular component excluded) is a measurement mode that excludes specular reflection light. The measurement model of SCE was adopted in this study. Finally, three standard surfaces are given: metal surface ball A, ceramics surface (black) ball B, and ceramics surface(white) ball C. The reflectance of the standard spheres of the three different materials are shown in Table 1. By measuring the reflectance, we obtained the reflectance of the metal standard sphere (A) and black ceramic sphere (B) with similar reflectance, and the reflectance of the white ceramic sphere (C) with relatively high reflectance. The experimental results can be used to compare different materials with similar reflectance and the same material with a large difference in reflectance.

### 3.2. Calibration Experiment of Ceramic Standard Spheres Scanning with Large Reflectance Difference

In order to investigate the effect of surface reflectance on scanning measurement, scanning tests were carried out to compare the black standard ceramic sphere and white standard ceramic sphere, respectively. The point scanning system (MV330 LASER RADAR) with the maximum permissible error (MPE) ±0.3 mm was used for scanning; two ceramic balls were scanned simultaneously under the same conditions to obtain the surface morphologies.

The test samples are chosen as the black standard ceramic spheres (ball B) and white ceramic balls (ball C), respectively, as shown in Figure 4 and Figure 5. The distance between the ceramic ball and the scanning system is approximately 3 m.

Figure 4b presents the measured results of the black standard ceramic sphere. The point cloud is shown using the best fitting sphere [29]. The fitting condition is that the maximum angle is 45°. The ratio of eliminating the noise is 5% and the same parameters were further employed in the following scanning tests. In the experiment, 25,298 points are obtained. According to the measured result, the fitted diameter is 101.488 mm, and the root mean square (RMS) is 0.013 mm. The experimental results indicate that the measured surface is relatively smooth and the surface roughness is within the accuracy of the instrument, as shown in Figure 4b.

Since the object to be measured is a sphere, the scanning system must have almost vertical incidence, or else for the measured ball with high reflectance, invalid data will appear. It can be seen from Figure 5b,c that when the incident angle is close to zero, some remarkable protrusions appear on the spherical surface, which is due to localized high reflectance. Based on the obtained data cloud (24,927 points), the fitting calculation diameter is 99.746 mm, and the RMS is 0.094 mm. From the measurement results, the RMS value of ball B is 0.013 mm, and the diameter measurement difference is −0.112 mm. Under the same measurement conditions, the RMS value of ball C is 0.094 mm, and the diameter measurement difference is −0.254 mm. During the experiment, we took many measurements, and the trend of the results is consistent with the above results. Namely, the measurement deviation and RMS value of the measured white ceramic sphere with high reflectance are much higher than that of the black ceramic sphere with low reflectance.

### 3.3. Standard Spheres Scanning Calibration Experiments for Different Materials and Reflectance

In order to verify the influence of different materials on the scanning calibration results, three calibrated standard spheres were scanned simultaneously. As shown in Figure 6, the diameter of ball A is 101.600 mm, the diameter of ball B is 101.600 mm, and the diameter of ball C is 100.000 mm.

A LIDAR scanner (MV330) is used to scan the surfaces of the three balls at approximately equal distances. The detailed parameters of the MV330 are as follows: Principle: frequency modulated continuous wave coherent laser radar (FMCW laser radar). Wavelength: 1550 nm. Azimuth: ±360°. Elevation: ±45°. Range: 1~30 m. Range uncertainty Ur = 10 μm + 2.5 μm/(k = 2). Azimuth uncertainty Uaz = 0.7 arc-sec (k = 2). Elevation uncertainty Uel = 0.7 arc-sec (k = 2). Combined coordinate error, MPE = 20 μm + 19 μm/m

The obtained point clouds for the three balls are shown in Figure 7, and the re-constructed fitting spheres based on the three point clouds by the scanning are shown in Figure 8. From the scanning measurement data of different balls, as shown in Table 2, it can be seen that the RMS parameters of the white ceramic ball with a reflectance of approximately 60% are ten times higher than those of the metal ball with a reflectance of approximately 15% and the black ceramic ball with a reflectance of approximately 11%, and the point cloud data measured by the metal ball have the smallest RMS parameters in the ball fitting. The RMS parameter of the black ceramic ball with a reflectance of approximately 11% is of the same order of magnitude as that of the metal sphere, but the dimensional deviation is twice that of the metal ball.

Through the scanning calibration comparison experiment of a metal standard sphere with similar reflectance, black ceramic standard sphere, and white standard sphere with high reflectance, we find that there are slight differences in the calibration results when the materials are different and reflectance is similar. However, for the white standard sphere with 60% reflectance, the calibration result has an obviously large deviation.

### 3.4. Contrast Experiment of Plane Scanning

When measuring the spherical standard body, the laser scanner inevitably found a difference in incident angle. In order to exclude the possibility that the difference in the measurement results in the standard sphere calibration experiment was caused by the difference in incidence angle, we conducted a comparison experiment between the metal standard plane with a reflectance of 15% and the ceramic standard plane with a reflectance of 60% at a fixed incidence angle of 25–75°.

As shown in Figure 9, the metal plane with a matte surface and reflectance of approximately 15% is selected as the test sample, and the working surface size is approximately 460 mm × 190 mm with 0.2 mm flatness. In addition, another sample is used: a white ceramic plane with a matte surface and reflectance of approximately 60%. The working surface size is of approximately 450 × 80 mm with 0.01 mm flatness. The metal plane and white ceramic plane are 3 m away from the scanning system during scanning.

As shown in Figure 10, the measured surface is re-constructed by the obtained point cloud shown using the best fitting plane. The fitting flatness of the measured plane is 0.345 mm, with 138,681 points, and the standard deviation and RMS are both 0.077 mm. GD&T’s evaluation standard uses ASME Y14.5 2009.

As shown in Figure 11, the calculation method of the best fitting plane is used to calculate the point cloud of the white ceramic plane, which includes 1824 valid points. The purple plane is the plane formed by point cloud data, and the blue plane is the theoretical plane formed by point cloud data fitting. Based on the standard plane, the peak/valley value of the point cloud data reaches 11.134 mm. The standard deviation and RMS are 2.916 mm and 2.915 mm, respectively. The evaluation standard of GD&T is ASME Y14.5 2009.

As shown in Table 3, it can be concluded that high surface reflectance will adversely affect the measurement of both spherical and plane objects.

## 4. Conclusions

In order to investigate the effect of standard spheres or planes with different materials and reflectance on the non-contact scanning measurement results during instrument calibration, point scanning systems were used to analyze and compare the scanning data derived from metal, black and white ceramic balls, a white ceramic plane, and a gray metal plane, with different reflectance. The performance of the instrument was found to be related to the surface. The RMS of the white ceramic ball with a reflectance of approximately 60% was ten times higher than that of the metal ball with a reflectance of approximately 15%. It is necessary to provide the reflectivity of the standard in the calibration process of a 3D scanner, and we suggest that such requirements be given in the measurement standards. In the subsequent research and test work, we will test and evaluate aluminum materials widely used in the automotive and aerospace industries and we will combine other measurement methods such as radar, sonar, and fusion sensor to further study whether the measurement scheme proposed in this paper is applicable to other fields [30,31,32].

## Figures and Tables

**Figure 1 micromachines-13-01607-f001:**
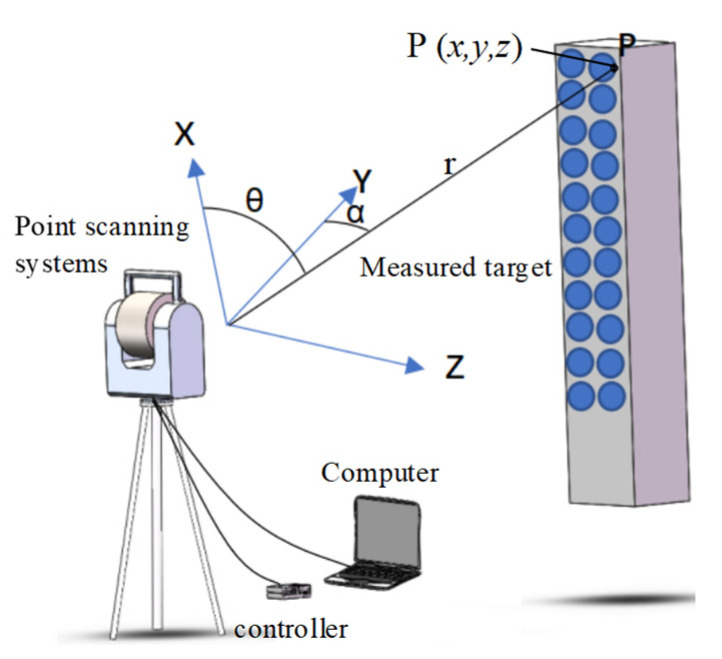
Schematic diagram of mechanism of single-point laser scanning measurement in spherical coordinate.

**Figure 2 micromachines-13-01607-f002:**
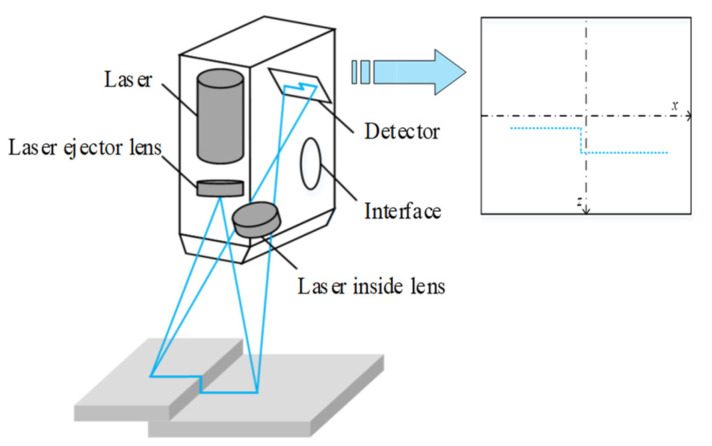
Schematic diagram of mechanism of single-line laser scanning system.

**Figure 3 micromachines-13-01607-f003:**
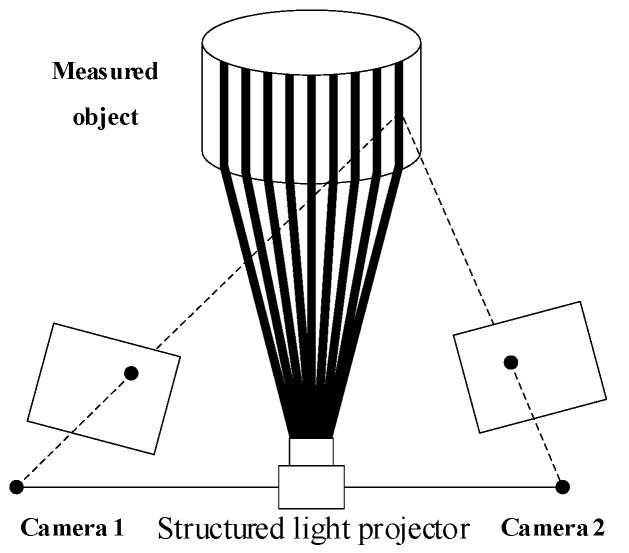
Schematic diagram of mechanism of stripe-area laser scanning system.

**Figure 4 micromachines-13-01607-f004:**
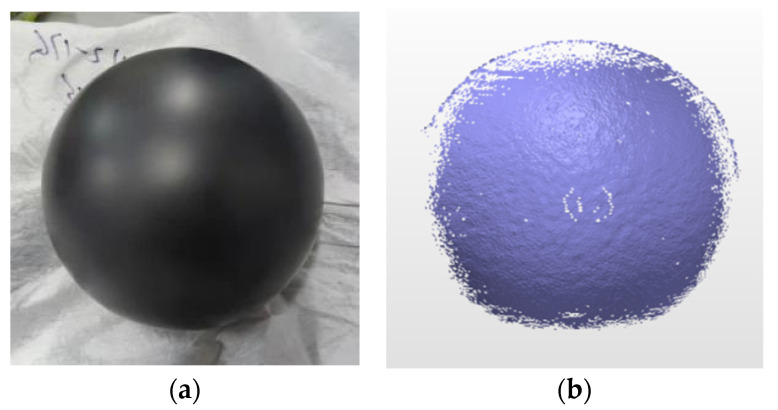
(**a**) Photograph of the black standard ceramic sphere with matte surface, and (**b**) point cloud obtained by point scanning system.

**Figure 5 micromachines-13-01607-f005:**
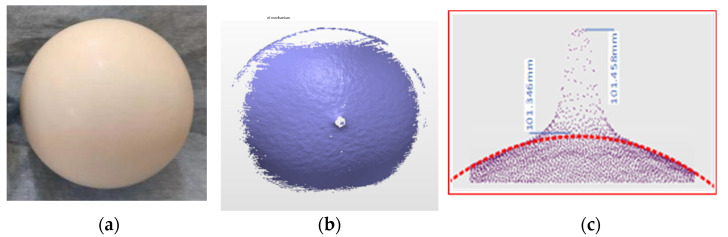
(**a**) White ceramic ball with bright surface with reflectance of approximately 90%, (**b**) point cloud data at vertical incidence, and (**c**) close view of the protrusion marked in (**b**).

**Figure 6 micromachines-13-01607-f006:**
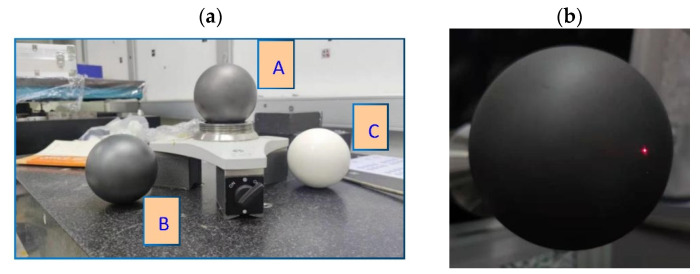
Scanning measurement comparison based on different reflectance: (**a**) The standard spheres made by three different materials with different reflectance, and (**b**) scanning light spot on black ceramic surface.

**Figure 7 micromachines-13-01607-f007:**
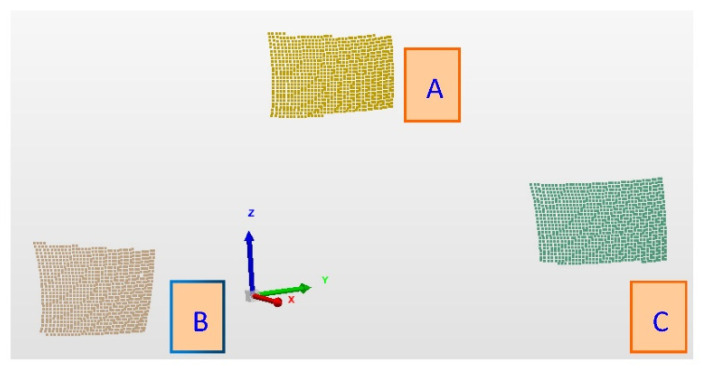
Point clouds obtained from laser scanning of the three balls.

**Figure 8 micromachines-13-01607-f008:**
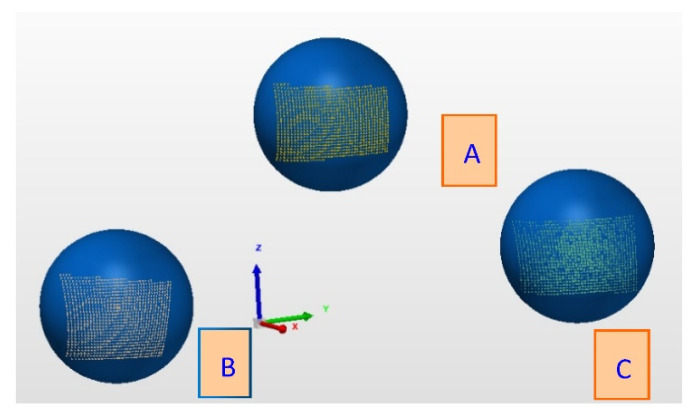
The re-construction based on the point clouds by the scanning measurements.

**Figure 9 micromachines-13-01607-f009:**
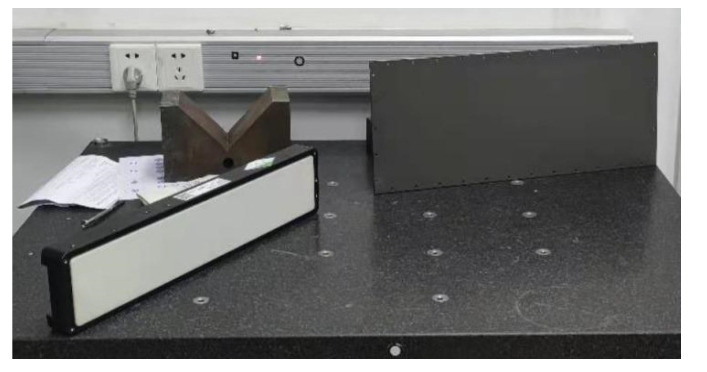
The photographs of the metal plane and white ceramics plane.

**Figure 10 micromachines-13-01607-f010:**
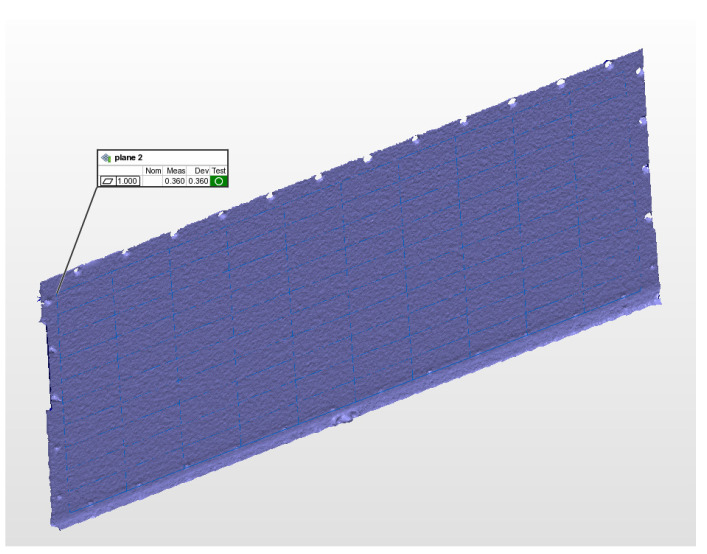
The re-construction of the metal plane based on the measured point cloud.

**Figure 11 micromachines-13-01607-f011:**
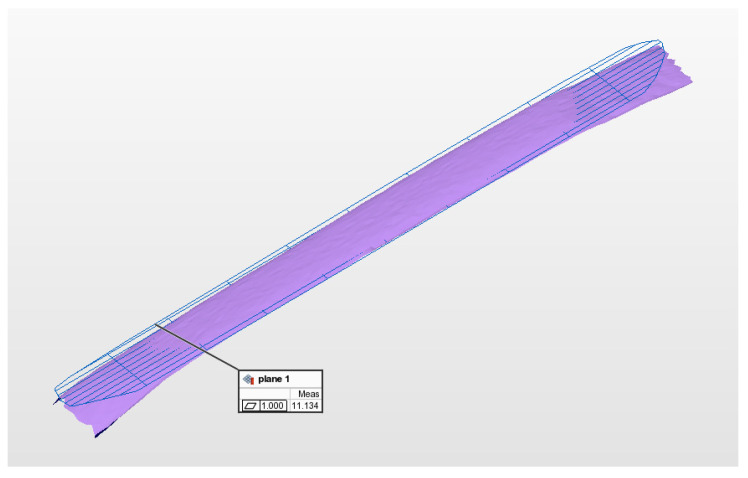
Triangular surface slice results of ceramic plane point cloud.

**Table 1 micromachines-13-01607-t001:** The reflectance of standard spheres made by three different materials.

No.	Materials	Surface Treatment	Surface Reflectance
1	Metal	Matte	15%
2	Ceramics (black)	Matte	11%
3	Ceramics (white)	Matte	60%

**Table 2 micromachines-13-01607-t002:** Comparison of LiDAR scanning results for a 101.6 mm ball.

Materials	Pts	Diameter	Deviation	RMS	RMS Fixed Radius
Metal	862	101.463	−0.137	0.009	0.011
Ceramics (black)	949	101.313	−0.287	0.013	0.019
Ceramics (white)	892	99.709	−0.291	0.099	0.102

**Table 3 micromachines-13-01607-t003:** Comparison of scanning results.

Materials	Surface Treatment	Surface Reflectance	RMS	RMS Fixed Radius
Metal	Matte	15%	0.009	0.011
Ceramics (black)	Matte	11%	0.013	0.019
Ceramics (white)	Matte	60%	0.099	0.102

## Data Availability

Not applicable.

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
