# Peer review of "Comparative Analysis on the Effect of Surface Reflectance for Laser 3D Scanner Calibrator"

_micromachines, 2022, doi:10.3390/mi13101607_

Round 1

Reviewer 1 Report

Analysis on the Effect of Surface Reflectance for 2 Laser 3D Scanner Calibrator:

Non-contact measurement technology is currently highly needed for automotive, agriculture and aerospace industries, good work!

I would recommend that you can also evaluate Aluminum, that will be very interesting for Automotive and aerospace industry.

Conclusions: RMS of ceramic spere is much higher than the metal sphere, is there any reference you can also rely based on your previous experience.

Good references, please review the format because some references are in Caps...

In section 2, can you describe more in detail about the Material's preparation or characteristics to be analyzed by this methodology/technique?

I understand you are working on the spheres as initial surfaces, what will happen for instance for metal components which have oil or cutting fluid in the surface? is this technique able to work on specimens like those?

Author Response

Reviewer: 1

  1. I would recommend that you can also evaluate Aluminum, that will be very interesting for Automotive and aerospace industry.

We have accepted the reviewer's suggestion to test and evaluate aluminum materials widely used in the automotive and aerospace industries in subsequent work. In addition, the following statement is added to the conclusion of the article (Line 328-331): In the subsequent research and test work, we will test and evaluate aluminum materials widely used in the automotive and aerospace industries and we will combine other measurement methods such as radar, sonar and fusion sensor to further study whether the measurement scheme proposed in this paper is applicable to other fields.

  1. Conclusions: RMS of ceramic spere is much higher than the metal sphere, is there any reference you can also rely based on your previous experience.

Our test in this paper is aimed at the effect of different materials and different surface treatment results on laser 3D scanning. The root mean square error (RMS) of ceramic ball is much higher than that of metal ball. This conclusion is based on practical experience and experimental results, and no relevant reference materials are retrieved.

  1. Good references, please review the format because some references are in Caps...

We have accepted the reviewer's comments and corrected 1, 4, 8, 9, 10, 11, 22, 28, 29 in the references section according to the journal template.

  1. In section 2, can you describe more in detail about the Material's preparation or characteristics to be analyzed by this methodology/technique?

 I understand you are working on the spheres as initial surfaces, what will happen for instance for metal components which have oil or cutting fluid in the surface? is this technique able to work on specimens like those?

We have accepted the comments of the reviewer. This technical study proposed in this paper is a method study for the measurement calibration process of instruments and equipment, so it is necessary to clean up the surface to reduce interference factors. In the article (Line 170-172), the following content is added: Before the test is carried out, we need to clean the surface of the standard instrument under test. The accuracy of the measurement results is ensured by removing surface oil or cutting fluid residue through cleaning operation.

Reviewer 2 Report

This article presents a comparative analysis on the effect of surface reflectance for 2 Laser 3D scanner calibrator. The main conclusion is that RMS of the flatness measurement is 0.077 mm for the metal plane calibrator 22 with a reflectance of 15%, and 2.915 mm for ceramic plane calibrator with a reflectance of 60%. In general, the article is well organized with theoretical analysis and experimental verifications presented. The experimental results are interesting revealing the effectiveness of the method, while some concerns need to be addressed, which are listed as below:

(1) The novelties and technical contributions need to be highlighted, current version looks like an experimental report. The theoretical contributions can also be presented more clearly.

(2) The quantitative results revealing the principle of calibration need to be provided.

(3) The comparison with other analysis methods is suggested to be added. It is suggested to discuss whether the proposed method can be adapted to other measurement methods such as radar, sonar, fusion sensor, etc.

(4) There are some chip-level LiDAR, radar, and fusion sensing solutions for wide FoV scanning and accurate detection on the surface by calibration. It would be better to discuss whether the proposed method be expanded and extended for radar or LiDAR calibrations for accurate wide FoV sensing. Some reference works for the possible information are listed as below:

(i) L. Lou et al., "14.2 An Early Fusion Complementary RADAR-LiDAR TRX in 65nm CMOS Supporting Gear-Shifting Sub-cm Resolution for Smart Sensing and Imaging," 2021 IEEE International Solid- State Circuits Conference (ISSCC), 2021, pp. 220-222, doi: 10.1109/ISSCC42613.2021.9365756.

(ii) Z. Fang et al., "Wide Field-of-View Locating and Multimodal Vital Sign Monitoring Based on X-Band CMOS-Integrated Phased-Array Radar Sensor," IEEE Transactions on Microwave Theory and Techniques, vol. 68, no. 9, pp. 4054-4065, Sept. 2020, doi: 10.1109/TMTT.2020.2989284.

(iii) Z. Fang et al., "A Silicon-Based Adaptable Edge Coherent Radar Platform for Seamless Health Sensing and Cognitive Interactions With Human Subjects," IEEE Transactions on Biomedical Circuits and Systems, vol. 16, no. 1, pp. 138-152, Feb. 2022, doi: 10.1109/TBCAS.2022.3145861.

Author Response

Reviewer: 2

  1. The novelties and technical contributions need to be highlighted, current version looks like an experimental report. The theoretical contributions can also be presented more clearly.

We have accepted the comments of the reviewer. In addition, the following statement is added in this paper (Line 94-97): This article first puts forward different reflectivity of the same 3d scanning device scans have different effect, this conclusion can not only improve existing calibration specifications, can also for other scanning measurement calibration equipment research and development and to provide the reference.

  1. The quantitative results revealing the principle of calibration need to be provided.

We have accepted the comments of the reviewer. In addition, the following statement is added in this paper (Line 314):

Table 3. Comparison of scanning results.

Materials

Surface treatment

Surface reflectance

RMS

RMS

Fixed radius

Metal

Matte

15%

0.009

0.011

Ceramics (black)

Matte

11%

0.013

0.019

Ceramics (white)

Matte

60%

0.099

0.102

  1. The comparison with other analysis methods is suggested to be added. It is suggested to discuss whether the proposed method can be adapted to other measurement methods such as radar, sonar, fusion sensor, etc.

We have accepted the reviewer's comments and will study whether the method is applicable to radar, sonar and fusion sensors in the subsequent work. In addition, the following statement is added in the conclusion of this paper (Line 328-331): In the subsequent research and test work, we will test and evaluate aluminum materials widely used in the automotive and aerospace industries and we will combine other measurement methods such as radar, sonar and fusion sensor to further study whether the measurement scheme proposed in this paper is applicable to other fields.

  1. There are some chip-level LiDAR, radar, and fusion sensing solutions for wide FoV scanning and accurate detection on the surface by calibration. It would be better to discuss whether the proposed method be expanded and extended for radar or LiDAR calibrations for accurate wide FoV sensing. Some reference works for the possible information are listed as below:

(i) L. Lou et al., "14.2 An Early Fusion Complementary RADAR-LiDAR TRX in 65nm CMOS Supporting Gear-Shifting Sub-cm Resolution for Smart Sensing and Imaging," 2021 IEEE International Solid- State Circuits Conference (ISSCC), 2021, pp. 220-222, doi: 10.1109/ISSCC42613.2021.9365756.

(ii) Z. Fang et al., "Wide Field-of-View Locating and Multimodal Vital Sign Monitoring Based on X-Band CMOS-Integrated Phased-Array Radar Sensor," IEEE Transactions on Microwave Theory and Techniques, vol. 68, no. 9, pp. 4054-4065, Sept. 2020, doi: 10.1109/TMTT.2020.2989284.

(iii) Z. Fang et al., "A Silicon-Based Adaptable Edge Coherent Radar Platform for Seamless Health Sensing and Cognitive Interactions With Human Subjects," IEEE Transactions on Biomedical Circuits and Systems, vol. 16, no. 1, pp. 138-152, Feb. 2022, doi: 10.1109/TBCAS.2022.3145861.

We have accepted the comments of the reviewer. The three articles provided by the reviewer have some contributions to the integrity of this paper. We have added the citations of the above three articles in the references section.